# PINNsFormer: A Transformer-Based Framework For Physics-Informed Neural Networks

**Zhiyuan Zhao**
Georgia Institute of Technology
Atlanta, GA 30332
`leozhao1997@gatech.edu`

**Xueying Ding**
Carnegie Mellon University
Pittsburgh, PA 15213
`xding2@andrew.cmu.edu`

**B. Aditya Prakash**
Georgia Institute of Technology
Atlanta, GA 30332
`baadityap@cc.gatech.edu`

## Abstract

Physics-Informed Neural Networks (PINNs) have emerged as a promising deep learning framework for approximating numerical solutions to partial differential equations (PDEs). However, conventional PINNs, relying on multilayer perceptrons (MLP), neglect the crucial temporal dependencies inherent in practical physics systems and thus fail to propagate the initial condition constraints globally and accurately capture the true solutions under various scenarios. In this paper, we introduce a novel Transformer-based framework, termed PINNsFormer, designed to address this limitation. PINNsFormer can accurately approximate PDE solutions by utilizing multi-head attention mechanisms to capture temporal dependencies. PINNsFormer transforms point-wise inputs into pseudo sequences and replaces point-wise PINNs loss with a sequential loss. Additionally, it incorporates a novel activation function, `Wavelet`, which anticipates Fourier decomposition through deep neural networks. Empirical results demonstrate that PINNsFormer achieves superior generalization ability and accuracy across various scenarios, including PINNs failure modes and high-dimensional PDEs. Moreover, PINNsFormer offers flexibility in integrating existing learning schemes for PINNs, further enhancing its performance.

## 1 Introduction

Numerically solving partial differential equations (PDEs) has been widely studied in science and engineering. The conventional approaches, such as finite element method (Bathe, 2007) or pseudo-spectral method (Fornberg, 1998), suffer from high computational costs in constructing meshes for high-dimensional PDEs. With the development of scientific machine learning, Physics-informed neural networks (PINNs) (Lagaris et al., 1998; Raissi et al., 2019) have emerged as a promising novel approach. Conventional PINNs and most variants employ multilayer perceptrons (MLP) as end-to-end frameworks for point-wise predictions, achieving remarkable success in various scenarios.

Nevertheless, recent works have shown that PINNs fail in scenarios when solutions exhibit high-frequency or multiscale features (Raissi, 2018; Fuks & Tchelepi, 2020; Krishnapriyan et al., 2021; Wang et al., 2022a), though the corresponding analytical solutions are simple. In such cases, PINNs tend to provide overly smooth or naive approximations, deviating from the true solution.

Existing approaches to mitigate these failures typically involve two general strategies. The first strategy, known as data interpolation (Raissi et al., 2017; Zhu et al., 2019; Chen et al., 2021), employs data regularization observed from simulations, or real-world scenarios. These approaches face challenges in acquiring ground truth data. The second strategy employs different training schemes (Mao et al., 2020; Krishnapriyan et al., 2021; Wang et al., 2021; 2022a), which potentially impose a high computational cost in practice. For instance, *Seq2Seq* by Krishnapriyan et al. (2021) requires training multiple neural networks sequentially, while other networks suffer from convergence issues

due to error accumulation. Another method, *Neural Tangent Kernel (NTK)* (Wang et al., 2022a), involves constructing kernels $K \in \mathbb{R}^{D \times P}$, where $D$ is the sample size and $P$ is the model parameter, which suffers from scalability issues as the sample size or model parameter increases.

While most efforts to improve the generalization ability and address failure modes in PINNs have focused on the aforementioned aspects, conventional PINNs, largely relying on MLP-based architecture, can overlook important temporal dependencies in real-world physical systems. Finite Element Methods, for instance, implicitly incorporate temporal dependencies by sequentially propagating the global solution. This propagation relies on the principle that the state at time $t + \Delta t$ depends on the state at time $t$. In contrast, PINNs, being a point-to-point framework, do not explicitly model temporal dependencies within PDEs. Neglecting temporal dependencies poses challenges in globally propagating initial condition constraints in PINNs. Consequently, PINNs often exhibit failure modes where the approximations remain accurate near the initial condition but subsequently fail into overly smooth or naive approximations.

To address this issue of neglecting temporal dependencies in PINNs, a natural idea is employing Transformer-based models, which are known for capturing long-term dependencies in sequential data through multi-head self-attentions and encoder-decoder attentions (Vaswani et al., 2017). Variants of transformer-based models have shown substantial success across various domains. However, adapting the Transformer, which is inherently designed for sequential data, to the point-to-point framework of PINNs presents non-trivial challenges. These challenges span both the data representation and the regularization loss within the framework.

**Main Contributions.** In this work, we introduce PINNsFormer, a novel sequence-to-sequence PDE solver built on the Transformer architecture. To the best of our knowledge, PINNsFormer is the first framework in the realm of PINNs that explicitly focuses on and learns temporal dependencies within PDEs. Our key contributions can be summarized as follows:

- **New Framework:** We propose a novel yet intuitive Transformer-based framework named PINNsFormer. This framework equips PINNs with the capability to capture temporal dependencies through the generated pseudo sequences, thereby enhancing the generalization ability and approximation accuracy in effectively solving PDEs.
- **Novel Activation:** We introduce a new non-linear activation function `Wavelet`. `Wavelet` is designed to anticipate the Fourier Transform for arbitrary target signals, making it a universal approximator for infinite-width neural networks. `Wavelet` can also be potentially beneficial to various deep learning tasks across different model architectures.
- **Extensive Experiments:** We conduct comprehensive evaluations of PINNsFormer for various scenarios. We demonstrate its advantages in optimization and approximation accuracy when addressing failure modes or solving high-dimensional PDEs. We show the flexibility and benefits of PINNsFormer in incorporating different learning schemes of PINNs.

## 2 RELATED WORK

**Physics-Informed Neural Networks (PINNs).** Physics-Informed Neural Networks (PINNs) have emerged as a promising approach for tackling scientific and engineering problems. Raissi et al. (2019) introduced the framework that incorporates physical laws into the neural network training to solve PDEs. This work has led to applications across diverse domains, including fluid dynamics, solid mechanics, and quantum mechanics (Carleo et al., 2019; Yang et al., 2020). Researchers have investigated different learning schemes for PINNs (Mao et al., 2020; Wang et al., 2021; 2022a), which have yielded substantial improvements in convergence, generalization, and interpretability.

**Failure Modes of PINNs.** Despite the promise exhibited by PINNs, recent works have indicated certain failure modes inherent to PINNs, particularly when confronted with PDEs featuring high-frequency or multiscale features (Fuks & Tchelepi, 2020; Raissi, 2018; McClenny & Braga-Neto, 2020; Krishnapriyan et al., 2021; Zhao et al., 2022; Wang et al., 2022a). This challenge has prompted investigations from various perspectives, including designing various model architectures, learning schemes, or using data interpolations (Han et al., 2018; Lou et al., 2021; Wang et al., 2021; 2022a;b). A comprehensive understanding of PINNs' limitations and the underlying failure modes is fundamental for applications in addressing complicated physical problems.

Figure 1: Architecture of proposed PINNsFormer. PINNsFormer generates a pseudo sequence based on pointwise input features. It outputs the corresponding sequential approximated solution. The first approximation of the sequence is the desired solution $\hat{u}(\boldsymbol{x}, t)$.

**Transformer-Based Models.** The Transformer model (Vaswani et al., 2017) has achieved significant attention due to its ability to capture long-term dependencies, leading to major achievements in natural language processing tasks (Devlin et al., 2018; Radford et al., 2018). Transformers have also been extended to other domains, including computer vision, speech recognition, and time-series analysis (Liu et al., 2021; Dosovitskiy et al., 2020; Gulati et al., 2020; Zhou et al., 2021). Researchers have also developed techniques aimed at enhancing the efficiency of Transformers, such as sparse attention and model compression (Child et al., 2019; Sanh et al., 2019).

## 3 METHODOLOGY

**Preliminaries:** Let $\Omega$ be an open set in $\mathbb{R}^d$, bounded by $\partial\Omega \in \mathbb{R}^{d-1}$. The PDEs with spatial input $\boldsymbol{x}$ and temporal input $t$ generally fit the following abstraction:

$$\begin{aligned} \mathcal{D}[u(\boldsymbol{x}, t)] &= f(\boldsymbol{x}, t), \ \ \forall \boldsymbol{x}, t \in \Omega \\ \mathcal{B}[u(\boldsymbol{x}, t)] &= g(\boldsymbol{x}, t), \ \ \forall \boldsymbol{x}, t \in \partial\Omega \end{aligned} \quad (1)$$

where $u$ is the PDE's solution, $\mathcal{D}$ is the differential operator that regularizes the behavior of the system, and $\mathcal{B}$ describes the boundary or initial conditions in general. Specifically, $\{\boldsymbol{x}, t\} \in \Omega$ are residual points, and $\{\boldsymbol{x}, t\} \in \partial\Omega$ are boundary/initial points. Let $\hat{u}$ be neural network approximations, PINNs describe the framework where $\hat{u}$ is empirically regularized by the following constraints:

$$\mathcal{L}_{\texttt{PINNs}} = \lambda_r \sum_{i=1}^{N_r} \|\mathcal{D}[\hat{u}(\boldsymbol{x}, t)] - f(\boldsymbol{x}, t)\|^2 + \lambda_b \sum_{i=1}^{N_b} \|\mathcal{B}[\hat{u}(\boldsymbol{x}, t)] - g(\boldsymbol{x}, t)\|^2 \quad (2)$$

where $N_b, N_r$ refer to the residual and boundary/initial points separately, $\lambda_r, \lambda_b$ are the regularization parameters that balance the emphasis of the loss terms. The neural network $\hat{u}$ takes vectorized $\{\boldsymbol{x}, t\}$ as input and outputs the approximated solution. The goal is then to use machine learning methodologies to train the neural network $\hat{u}$ that minimizes the loss in Equation 2.

**Methodology Overview:** While PINNs focus on point-to-point predictions, the exploration of temporal dependencies in real-world physics systems has been merely neglected. Conventional PINNs methods employ a single pair of spatial information $\boldsymbol{x}$ and temporal information $t$ to approximate the numerical solution $u(\boldsymbol{x}, t)$, without accounting for temporal dependencies across previous or subsequent time steps. However, this simplification is only applicable to elliptic PDEs, where the relationships between unknown functions and their derivatives do not explicitly involve time. In contrast, hyperbolic and parabolic PDEs incorporate time derivatives, implying that the state at one time step can influence states at preceding or subsequent time steps. Consequently, considering temporal dependencies is crucial to effectively address these PDEs using PINNs.

In this section, we introduce a novel framework featuring a Transformer-based model of PINNs, namely PINNsFormer. Unlike point-to-point predictions, PINNsFormer extends PINNs' capabilities to sequential predictions. PINNsFormer allows accurately approximating solutions at specific time steps while also learning and regularizing temporal dependencies among incoming states. The framework consists of four components: Pseudo Sequence Generator, Spatio-Temporal Mixer, Encoder-Decoder with multi-head attention, and an Output Layer. Additionally, we introduce a novel activation function, named `Wavelet`, which employs Real Fourier Transform techniques to anticipate solutions to PDEs. The framework diagram is exhibited in Figure 1. We provide detailed explanations of each framework component and learning schemes in the following subsections.

## 3.1 PSEUDO SEQUENCE GENERATOR

While Transformers and Transformer-based models are designed to capture long-term dependencies in sequential data, conventional PINNs utilize non-sequential data as inputs for neural networks. Consequently, to incorporate PINNs with Transformer-based models, it is essential to transform the pointwise spatiotemporal inputs into temporal sequences. Thus, for a given spatial input $\boldsymbol{x} \in \mathbb{R}^{d-1}$ and temporal input $t \in \mathbb{R}$, the Pseudo Sequence Generator performs the following operations:

$$[\boldsymbol{x}, t] \xrightarrow{\text{Generator}} \{[\boldsymbol{x}, t], [\boldsymbol{x}, t + \Delta t], \ldots, [\boldsymbol{x}, t + (k-1)\Delta t]\} \tag{3}$$

where $[\cdot]$ is the concatenation operation, such that $[\boldsymbol{x}, t] \in \mathbb{R}^d$ is vectorized, and the generator outputs the pseudo sequence in the shape of $\mathbb{R}^{k \times d}$. The Pseudo Sequence Generator extrapolates sequential time series by extending a single spatiotemporal input to multiple isometric discrete time steps. $k$ and $\Delta t$ are hyperparameters, which intuitively determine how many steps the pseudo sequence needs to 'look ahead' and how 'far' each step should be. In practice, both $k$ and $\Delta t$ should not be set to very large scales, as larger $k$ can cause heavy computational and memory overheads, while larger $\Delta t$ may undermine the time dependency relationships of neighboring discrete time steps.

## 3.2 MODEL ARCHITECTURE

In addition to the Pseudo Sequence Generator, PINNsFormer consists of three components of its architecture: Sptio-Temporal Mixer, Encoder-Decoder with multi-head attentions, and Output Layer. The Output Layer is straightforward to interpret as a fully-connected MLP appended to the end. We provide detailed insights into the first two components below. Notably, PINNsFormer relies only on linear layers and non-linear activations, avoiding complex operations such as convolutional or recurrent layers. This design preserves PINNsFormer's computational efficiency in practice.

**Spatio-Temporal Mixer.** Most PDEs contain low-dimensional spatial or temporal information. Directly feeding low-dimensional data to encoders may fail to capture the complex relationships between each feature dimension. Hence, it is necessary to embed original sequential data in higher-dimensional spaces such that more information is encoded into each vector.

Instead of embedding raw data in a high-dimensional space where the distance between vectors reflects the semantic similarity (Vaswani et al., 2017; Devlin et al., 2018), PINNsFormer constructs a linear projection that maps spatiotemporal inputs onto a higher-dimensional space using a fully-connected MLP. The embedded data enriches the capability of information by mixing all raw spatiotemporal features together, so-called the linear projection Spatio-Temporal Mixer.

**Encoder-Decoder Architecture.** PINNsFormer employs an encoder-decoder architecture similar to Transformer. The encoder consists of a stack of identical layers, each of which contains an encoder self-attention layer and a feedforward layer. The decoder is slightly different from the vanilla Transformer, where each of the identical layers contains only an encoder-decoder self-attention layer and a feedforward layer. At the decoder level, PINNsFormer uses the same spatiotemporal embeddings as the encoder. Therefore, the decoder does not need to relearn dependencies for the same input embeddings. The diagram for the encoder-decoder architecture is shown in Figure 2

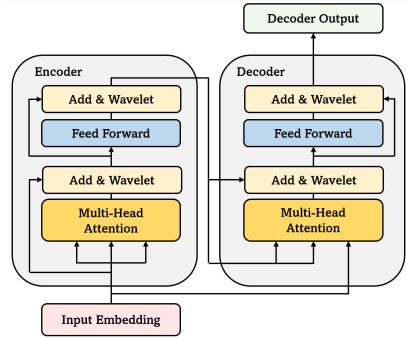

Figure 2: The architecture of PINNs-Former's Encoder-Decoder Layers. The decoder is not equipped with self-attentions.

Intuitively, the encoder self-attentions allow learning the dependency relationships of all spatiotemporal information. The decoder encoder-decoder attentions allow selectively focusing on specific dependencies within the input sequence during the decoding process, enabling it to capture more information than conventional PINNs. We use the same embeddings for the encoder and decoder since PINNs focus on approximating the solution of the *current* state, in contrast to *next* state prediction in language tasks or time series forecastings.

### 3.3 WAVELET ACTIVATION

While Transformers typically employ `LayerNorm` and `ReLU` non-linear activation functions (Vaswani et al., 2017; Gehring et al., 2017; Devlin et al., 2018), these activation functions might not always be suitable in solving PINNs. In particular, employing `ReLU` activation in PINNs can result in poor performance, whose effectiveness relies heavily on the accurate evaluation of derivatives while `ReLU` has a discontinuous derivative (Haghighat et al., 2021; de Wolff et al., 2021). Recent studies utilize `Sin` activation for specific scenarios to mimic the periodic properties of PDEs' solutions (Li et al., 2020; Jagtap et al., 2020; Song et al., 2022). However, it requires strong prior knowledge of the solution's behavior and is limited in its applicability. Tackling this issue, we proposed a novel and simple activation function, namely `Wavelet`, defined as follows:

$$\texttt{Wavelet}(\boldsymbol{x}) = \omega_1 \sin(\boldsymbol{x}) + \omega_2 \cos(\boldsymbol{x}) \tag{4}$$

Where $\omega_1$ and $\omega_2$ are registered learnable parameters. The intuition behind `Wavelet` activation simply follows Real Fourier Transform: While periodic signals can be decomposed into an integral of sines of multiple frequencies, all signals, whether periodic or aperiodic, can be decomposed into an integral of sines and cosines of varying frequencies. It is evident that `Wavelet` can approximate arbitrary functions giving sufficient approximation power, which leads to the following proposition:

**Proposition 1** *Let $\mathcal{N}$ be a two-hidden-layer neural network with infinite width, equipped with* `Wavelet` *activation function, then $\mathcal{N}$ is a universal approximator for any real-valued target f.*

*Proof sketch:* The proof follows the Real Fourier Transform (Fourier Integral Transform). For any given input $x$ and its corresponding real-valued target $f(x)$, it has the Fourier Integral:

$$f(x) = \int_{-\infty}^{\infty} F_c(\omega) \cos(\omega x)\, d\omega + \int_{-\infty}^{\infty} F_s(\omega) \sin(\omega x)\, d\omega$$

where $F_c$ and $F_s$ are the coefficients of Sines and Cosines respectively. Second, by Riemann sum approximation, the integral can be approximated by the infinite sum such that:

$$f(x) \approx \sum_{n=1}^{N} [F_c(\omega_n) \cos(\omega_n x) + F_s(\omega_n) \sin(\omega_n x)] \equiv W_2(\texttt{Wavelet}(W_1 x))$$

where $W_1$ and $W_2$ are the weights of $\mathcal{N}$'s first and second hidden layer. As $W_1$ and $W_2$ are infinite-width, we can divide the piecewise summation into infinitely small intervals, making the approximation arbitrarily close to the true integral. Hence, $\mathcal{N}$ is a universal approximator for any given $f$. In practice, most PDE solutions contain only a finite number of major frequencies. Using a neural network with finite parameters would also lead to proper approximations of the true solutions.

Although `Wavelet` activation function is primarily employed by PINNsFormer to improve PINNs in our work, it may have potential applications in other deep-learning tasks. Similar to `ReLU`, $\sigma(\cdot)$, and `Tanh` activations, which all turn infinite-width two-hidden-layer neural networks into universal approximators (Cybenko, 1989; Hornik, 1991; Glorot et al., 2011), we anticipate that `Wavelet` can demonstrate its effectiveness in other applications beyond the scope of this work.

### 3.4 LEARNING SCHEME

While conventional PINNs focus on point-to-point predictions, adapting PINNs to handle pseudo-sequential inputs has not been explored. In PINNsFormer, each generated point in the sequence, i.e., $[\boldsymbol{x}_i, t_i + j\Delta t]$, is mapped to the corresponding approximation, i.e., $\hat{u}(\boldsymbol{x}_i, t_i + j\Delta t)$ for any $j \in \mathbb{N}, j < k$. This approach allows us to compute the $n$th-order gradients with respect to $\boldsymbol{x}$ or $t$ independently for any valid $n$. For instance, for any given input pseudo sequence $\{[\boldsymbol{x}_i, t_i], [\boldsymbol{x}_i, t_i + \Delta t], \ldots, [\boldsymbol{x}_i, t_i + (k-1)\Delta t]\}$, and the corresponding approximations $\{\hat{u}(\boldsymbol{x}_i, t_i), \hat{u}(\boldsymbol{x}_i, t_i + \Delta t), \ldots, \hat{u}(\boldsymbol{x}_i, t_i + (k-1)\Delta t)\}$, we can compute the first-order derivatives w.r.t. $\boldsymbol{x}$ and $t$ separately as follows:

$$\begin{aligned} \frac{\partial \{\hat{u}(\boldsymbol{x}_i, t_i + j\Delta t)\}_{j=0}^{k-1}}{\partial \{t_i + j\Delta t\}_{j=0}^{k-1}} &= \{\frac{\partial \hat{u}(\boldsymbol{x}_i, t_i)}{\partial t_i}, \frac{\partial \hat{u}(\boldsymbol{x}_i, t_i + \Delta t)}{\partial (t_i + \Delta t)}, \ldots, \frac{\partial \hat{u}(\boldsymbol{x}_i, t_i + (k-1)\Delta t)}{\partial (t_i + (k-1)\Delta t)}\} \\ \frac{\partial \{\hat{u}(\boldsymbol{x}_i, t_i + j\Delta t)\}_{j=0}^{k-1}}{\partial \boldsymbol{x}_i} &= \{\frac{\partial \hat{u}(\boldsymbol{x}_i, t_i)}{\partial \boldsymbol{x}_i}, \frac{\partial \hat{u}(\boldsymbol{x}_i, t_i + \Delta t)}{\partial \boldsymbol{x}_i}, \ldots, \frac{\partial \hat{u}(\boldsymbol{x}_i, t_i + (k-1)\Delta t)}{\partial \boldsymbol{x}_i}\} \end{aligned} \tag{5}$$

This scheme for calculating the gradients of sequential approximations with respect to sequential inputs can be easily extended to higher-order derivatives and is applicable to residual, boundary, and initial points. However, unlike the general PINNs optimization objective in Equation 2, which combines initial and boundary condition objectives, PINNsFormer distinguishes between the two and applies different regularization schemes to initial and boundary conditions through its learning scheme. For residual and boundary points, all sequential outputs can be regularized using the PINNs loss. This is because all generated pseudo-timesteps are within the same domain as their original inputs. For example, if $[\boldsymbol{x}_i, t_i]$ is sampled from the boundary, then $[\boldsymbol{x}_i, t_i + j\Delta t]$ also lies on the boundary for any $j \in \mathbb{N}^+$. In contrast, for initial points, only the $t = 0$ condition is regularized, corresponding to the first element of the sequential outputs. This is because only the first element of the pseudo-sequence exactly matches the initial condition at $t = 0$. All other generated time steps have $t = j\Delta t$ for any $j \in \mathbb{N}^+$, which fall outside the initial conditions.

By these considerations, we adapt the PINNs loss to the sequential version, as described below:

$$\mathcal{L}_{res} = \frac{1}{kN_{res}} \sum_{i=1}^{N_{res}} \sum_{j=0}^{k-1} \|\mathcal{D}[\hat{u}(\boldsymbol{x}_i, t_i + j\Delta t)] - f(\boldsymbol{x}_i, t_i + j\Delta t)\|^2$$

$$\mathcal{L}_{bc} = \frac{1}{kN_{bc}} \sum_{i=1}^{N_{bc}} \sum_{j=0}^{k-1} \|\mathcal{B}[\hat{u}(\boldsymbol{x}_i, t_i + j\Delta t)] - g(\boldsymbol{x}_i, t_i + j\Delta t)\|^2 \quad (6)$$

$$\mathcal{L}_{ic} = \frac{1}{N_{ic}} \sum_{i=1}^{N_{bc}} \|\mathcal{I}[\hat{u}(\boldsymbol{x}_i, 0)] - h(\boldsymbol{x}_i, 0)\|^2$$

$$\mathcal{L}_{\texttt{PINNsFormer}} = \lambda_{res}\mathcal{L}_{res} + \lambda_{ic}\mathcal{L}_{ic} + \lambda_{bc}\mathcal{L}_{bc}$$

where $N_{res} = N_r$ refers to the residual points as in Equation 2, $N_{bc}, N_{ic}$ represent the number of boundary and initial points, respectively, with $N_{bc} + N_{ic} = N_b$. $\lambda_{res}, \lambda_{bc}$, and $\lambda_{ic}$ are regularization weights that balance the importance of the loss terms in PINNsFormer, similar to the PINNs loss.

During training, PINNsFormer forwards all residual, boundary, and initial points to obtain their corresponding sequential approximations. It then optimizes the modified PINNs loss $\mathcal{L}_{\texttt{PINNsFormer}}$ in Equation 6 using gradient-based optimization algorithms such as L-BFGS or Adam, updating the model parameters until convergence. In the testing phase, PINNsFormer forwards any arbitrary pair $[\boldsymbol{x}, t]$ to observe the sequential approximations, where the first element of the sequential approximation corresponds exactly to the desired value of $\hat{u}(\boldsymbol{x}, t)$.

## 3.5 LOSS LANDSCAPE ANALYSIS

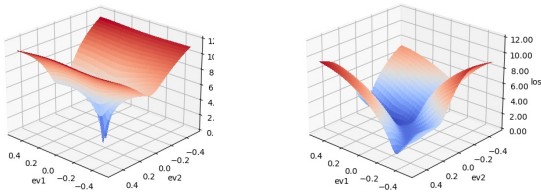

Figure 3: Visualization of the loss landscape for PINNs (left) and PINNsFormer (right) on a logarithmic scale. The loss landscape of PINNsFormer is significantly smoother than conventional PINNs.

While achieving theoretical convergence or establishing generalization bounds for Transformer-based models can be challenging, an alternative approach to assess optimization trajectory is through visualization of the loss landscape. This approach has been employed in the analysis of both Transformers and PINNs (Krishnapriyan et al., 2021; Yao et al., 2020; Park & Kim, 2022). The loss landscape is constructed by perturbing the trained model along the directions of the first two dominant Hessian eigenvectors. This technique is more informative than random parameter perturbations. Generally, a smoother loss landscape with fewer local minima indicates an easier convergence to the global minimum. We visualize the loss landscape for both PINNs and PINNsFormer. The visualizations are presented in Figure 5.

The visualizations clearly reveal that PINNs exhibit a more complicated loss landscape than PINNsFormer. To be specific, we estimate the Lipschitz constant for both loss landscapes. We find that $L_{\texttt{PINNs}} = 776.16$, which is significantly larger than $L_{\texttt{PINNsFormer}} = 32.79$. Furthermore, the loss

landscape of PINNs exhibits several sharp cones near its optimal point, indicating the presence of multiple local minima in close proximity to the convergence point (zero perturbation). The rugged loss landscape and multiple local minima of conventional PINNs suggest that optimizing the objective described in Equation 6 for PINNsFormer offers an easier path to reach the global minimum. This implies that PINNsFormer has advantages in avoiding the failure modes associated with PINNs. The analysis is further validated by empirical experiments, as shown in the following section.

## 4    EXPERIMENTS

### 4.1    SETUP

**Goal.** Our empirical evaluations aim to demonstrate three key advantages of PINNsFormer. First, we show that PINNsFormer improves generalization abilities and mitigates failure modes compared to PINNs and variant architectures. Second, we illustrate the flexibility of PINNsFormer in incorporating various learning schemes, resulting in superior performance. Third, we provide evidence of PINNsFormer's faster convergence and improved generalization capabilities in solving high-dimensional PDEs, which can be challenging for PINNs and their variants.

**Experiment Setup.** Our empirical evaluations rely on four types of PDEs: convection, 1D-reaction, 1D-wave, and Navier–Stokes PDEs, which follow the established setups of preliminary studies for fair comparisons (Raissi et al., 2019; Krishnapriyan et al., 2021; Wang et al., 2022a). We include PINNs, QRes (Bu & Karpatne, 2021), and First-Layer Sine (FLS) (Wong et al., 2022) as baselines. For convection, 1D-reaction, and 1D-wave PDEs, we uniformly sampled $N_{ic} = N_{bc} = 101$ initial and boundary points, as well as a uniform grid of $101 \times 101$ mesh points for the residual domain, resulting in total $N_{res} = 10201$ points. In the case of training PINNsFormer, we reduce the collocation points, with $N_{ic} = N_{bc} = 51$ initial and boundary points and a $51 \times 51$ mesh for residual points. The reduction in fewer training samples serves two purposes: it enhances training efficiency and allows us to demonstrate the generalization capabilities of PINNsFormer with limited training data. For testing, we employed a $101 \times 101$ mesh within the residual domain. For the Navier–Stokes PDE, we sample 2500 points from the 3D mesh within the residual domain for training purposes. The evaluation was performed by testing the predicted pressure at the final time step $t = 20.0$.

**Evaluation.** For all baselines and PINNsformer, we maintain approximately close numbers of parameters across all models to highlight the advantages of PINNsFormer from its ability to capture temporal dependencies rather than relying solely on model overparameterization. We train all models using the L-BFGS optimizer with Strong Wolfe linear search for 1000 iterations. For simplicity, we set $\lambda_{res} = \lambda_{ic} = \lambda_{bc} = 1$ for the optimization objective in Equation 6. Detailed hyperparameters are provided in Appendix A. We also include an ablation study on activation functions and a hyperparameter sensitivity study on the choice of $\{k, \Delta t\}$ in Appendix C.

In terms of evaluation metrics, we adopted commonly used metrics in related works (Krishnapriyan et al., 2021; Raissi et al., 2019; McClenny & Braga-Neto, 2020), including the relative Mean Absolute Error (rMAE or relative $\ell_1$ error) and the relative Root Mean Square Error (rRMSE or relative $\ell_2$ error). The detailed formulations of the metrics are provided in Appendix A.

**Reproducibility.** All models are implemented in PyTorch (Paszke et al., 2019), and are trained separately on single NVIDIA Tesla V100 GPU. All code and demos are included and reproducible at: `https://github.com/AdityaLab/pinnsformer`.

### 4.2    MITIGATING FAILURE MODES OF PINNs

Our primary evaluation focuses on demonstrating the superior generalization ability of PINNs-Former in comparison to PINNs, particularly on PDEs that are known to challenge PINNs' generalization capabilities. We focus on solving two distinct types of PDEs: the convection equation and the 1D-reaction equation. These equations pose significant challenges for conventional MLP-based PINNs, often resulting in what is referred to as "PINNs failure modes" (Mojgani et al., 2022; Daw et al., 2022; Krishnapriyan et al., 2021). In these failure modes, optimization gets stuck in local minima, leading to overly smooth approximations that deviate from the true solutions.

The objective of our evaluation is to showcase the enhanced generalization capabilities of PINNs-Former when compared to standard PINNs and their variations, specifically in addressing PINNs' failure modes. The evaluation results are summarized in Table 1, with detailed PDE formulations provided in Appendix B. We showcase the prediction and absolute error plots of PINNs and PINNs-Former on convection equation in Figure 4, all prediction plots available in Appendix C.

| Model | Convection | | | 1D-Reaction | | |
|---|---|---|---|---|---|---|
| | Loss | rMAE | rRMSE | Loss | rMAE | rRMSE |
| PINNs | 0.016 | 0.778 | 0.840 | 0.199 | 0.982 | 0.981 |
| QRes | 0.015 | 0.746 | 0.816 | 0.199 | 0.979 | 0.977 |
| FLS | 0.012 | 0.674 | 0.771 | 0.199 | 0.984 | 0.985 |
| PINNsFormer | **3.7e-5** | **0.023** | **0.027** | **3.0e-6** | **0.015** | **0.030** |

Table 1: Results for solving convection and 1D-reaction equations. PINNsFormer consistently outperforms all baseline methods in terms of training loss, rMAE, and rRMSE.

The evaluation results demonstrate significant outperformance of PINNsFormer over all baselines for both scenarios. PINNsFormer achieves the lowest training loss and test errors, distinguishing PINNsFormer as the only approach capable of mitigating the failure modes. In contrast, all other baseline methods remain stuck at global minima and fail to optimize the objective loss effectively. These results show the clear advantages of PINNsFormer in terms of generalization ability and approximation accuracy when compared to conventional PINNs and existing variants.

The additional concern for PINNsFormer is its computational and memory overheads relative to PINNs. While MLP-based PINNs are known for efficiency, PINNsFormer, with Transformer-based architecture in handling sequential data, naturally incurs higher computational and memory costs. Nonetheless, our empirical evaluation indicates that the overhead is tolerable, benefitting from the reliance on only linear layers, avoiding complicated operators such as convolution or recurrent layers. For instance, when setting the pseudo-sequence length $k = 5$, we observe an approximate 2.92x computational cost and a 2.15x memory usage (detailed in Appendix A). These overheads are reasonable in exchange for the substantial performance improvements by PINNsFormer.

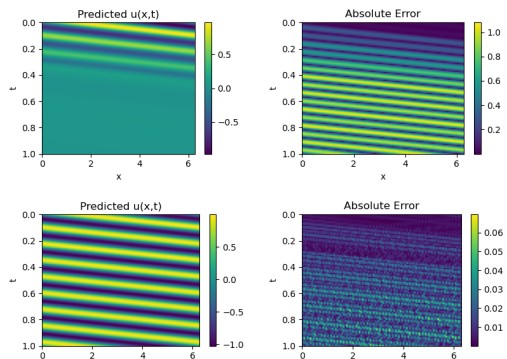

Figure 4: Prediction (left) and absolute error (right) of PINNs (up) and PINNsFormer (bottom) on convection equation. PINNsFormer shows success in mitigating the failure mode than PINNs.

## 4.3 FLEXIBILITY IN INCORPORATING VARIANT LEARNING SCHEMES

While PINNs and their various architectural adaptations may encounter challenges for certain scenarios, prior research has explored sophisticated optimization schemes to mitigate these issues, including learning rate annealing (Wang et al., 2021), augmented Lagrangian methods (Lu et al., 2021), and neural tangent kernel approaches (Wang et al., 2022a). These modified PINNs have shown significant improvement of PINNs under certain scenarios.

| Model | 1D-Wave | | |
|---|---|---|---|
| | Loss | rMAE | rRMSE |
| PINNs | 1.93e-2 | 0.326 | 0.335 |
| PINNsFormer | 1.38e-2 | 0.270 | 0.283 |
| PINNs + *NTK* | 6.34e-3 | 0.140 | 0.149 |
| PINNsFormer + *NTK* | **4.21e-3** | **0.054** | **0.058** |

Table 2: Results for solving the 1D-wave equation, incorporating the NTK method. PINNsFormer combined with NTK outperforms all other methods on all metrics.

Notably, when these optimization strategies are applied to PINNsFormer, they can be easily incorporated to achieve further performance improvements. For instance, the *Neural Tangent Kernel (NTK)* method to PINNs has shown success in solving the 1D-wave equation. As such, we demonstrate that when combining *NTK* with PINNsFormer, we can achieve

further outperformance in approximation accuracy. Detailed results are presented in Table 2, and comprehensive PDE formulations are available in Appendix B with prediction plots in Appendix C.

Our evaluation results show both the flexibility and effectiveness of incorporating PINNsFormer with the *NTK* method. In particular, we observe a sequence of performance improvements, from standard PINNs to PINNsFormer and from PINNs+*NTK* to PINNsFormer+*NTK*. Essentially, PINNsFormer explores a variant architecture of PINNs, while many learning schemes are designed from an optimization perspective and are agnostic to neural network architectures. This inherent flexibility allows for versatile combinations of PINNsFormer with various learning schemes, offering practical and customizable solutions for accurate solutions in real-world applications.

### 4.4 Generalization on High-Dimensional PDEs

In the previous sections, we demonstrated the clear benefits of PINNsFormer in generalizing the solutions for PINNs failure modes. However, those PDEs often have simple analytical solutions. In practical physics systems, higher-dimensional and more complex PDEs need to be solved. Therefore, it's important to evaluate the generalization ability of PINNsFormer on such high-dimensional PDEs, especially when PINNsFormer is equipped with advanced mechanisms like self-attention.

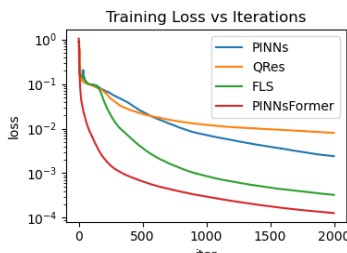

Figure 5: Training loss vs. Iterations of PINNs and PINNsFormer on the Navier-Stokes equation.

We evaluate the performance of PINNsFormer compared to PINNs on Navier-Stokes PDE based on the established setups Raissi et al. (2019). The training loss is shown in Figure 5, and the results are shown in Table 3. The detailed formulations of the 2D Navier-Stokes equation can be found in Appendix B, and the predictions are plotted in Appendix C.

The evaluation results demonstrate clear advantages of PINNsFormer over PINNs on high-dimensional PDEs. Firstly, PINNsFormer outperforms PINNs and their MLP-variants in terms of both training loss and validation errors. Firstly, PINNsFormer exhibits significantly faster convergence during training, which compensates for the higher computational cost per iteration. Secondly, while PINNs and their MLP-variants predict the pressure with good shapes, they exhibit increasing mag-

| Model | Navier-Stokes | | |
|---|---|---|---|
| | Loss | rMAE | rRMSE |
| PINNs | 6.72e-5 | 13.08 | 9.08 |
| QRes | 2.24e-4 | 6.41 | 4.45 |
| FLS | 9.54e-6 | 3.98 | 2.77 |
| PINNsFormer | **6.66e-6** | **0.384** | **0.280** |

Table 3: Results for solving Navier-Stokes equation, PINNsFormer outperforms all baselines on all metrics.

nitude discrepancies as time increases. In contrast, PINNsFormer consistently aligns both the shape and magnitude of predicted pressures across various time intervals. This consistency is attributed to PINNsFormer's ability to learn temporal dependencies through Transformer-based model architecture and self-attention mechanism.

## 5 Conclusion

In this paper, we introduced PINNsFormer, a novel Transformer-based framework of PINNs, aimed at capturing temporal dependencies when approximating solutions to PDEs. We introduced the Pseudo Sequence Generator, a mechanism that translates vectorized inputs into pseudo time sequences and incorporated a modified Encoder-Decoder layer along with a novel `Wavelet` activation. Empirical evaluations demonstrate that PINNsFormer consistently outperforms conventional PINNs across various scenarios, including handling PINNs' failure modes, addressing high-dimensional PDEs, and integrating with different learning schemes for PINNs. Furthermore, PINNsFormer retains computational simplicity, making it a practical choice for real-world applications.

Beyond PINNsFormer, `Wavelet` activation function can hold promises for the broader machine learning community. We provided a sketch proof demonstrating `Wavelet`'s ability to approximate arbitrary target solutions using a two-hidden-layer infinite-width neural network, leveraging the Fourier decomposition of these solutions. We encourage further exploration, both theoretically and empirically, of the `Wavelet` activation function's potential. Its applicability extends beyond PINNs and can be leveraged in various architectures and applications.

**Acknowledgements:** This paper was supported in part by the NSF (Expeditions CCF-1918770, CAREER IIS-2028586, Medium IIS-1955883, Medium IIS-2106961, PIPP CCF-2200269), CDC MInD program, Meta faculty gift, and funds/computing resources from Georgia Tech and GTRI.

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

## A    APPENDIX A: MODEL HYPERPARAMETERS

**Model Hyperparameters.** We provide a detailed set of hyperparameters used to obtain the experiment results, shown in Table 7.

| Model | Hyperparameters | Value | Model Parameters |
|---|---|---|---|
| PINNs & FLS | hidden layer | 4 | 527k |
|  | hidden size | 512 |  |
| QRes | hidden layer | 4 | 397k |
|  | hidden size | 256 |  |
| PINNsFormer | $k$ | 5 | 454k |
|  | $\Delta t$ | 1e-3, 1e-4 |  |
|  | # of encoder | 1 |  |
|  | # of decoder | 1 |  |
|  | embedding size | 32 |  |
|  | head | 2 |  |
|  | hidden size | 512 |  |

Table 4: Hyperparameters for Main Results

**Training Overhead.** We compare the training overhead of PINNsFormer over PINNs, as PINNs are known as an efficient framework while Transformer-based models are known for being computationally costly. The comparison relies on solving the Convection PDEs, which are detailed in Table 5. Here, we vary the hyperparameter of pseudo-sequence length $k$ for validation purposes. In practice, we set $k = 5$ for all the empirical experiments in this paper.

| Model | | Training Time (sec/epoch) | Computational Overhead | GPU Memory (MiB) | Memory Overhead |
|---|---|---|---|---|---|
| PINNs | | 0.80 | / | 1311 | / |
| PINNsFormer | $k$=3 | 2.10 | 2.62x | 2207 | 1.68x |
|  | $k$=5 | 2.34 | 2.92x | 2827 | 2.15x |
|  | $k$=10 | 3.10 | 3.87x | 4803 | 3.66x |

Table 5: Overhead of PINNsFormer than PINNs in varying pseudo-sequence length. Both computational and memory overhead are tolerable and grow approximately linearly as $k$ increases

**Evaluation Metrics.** We present the detailed formula of rMAE and rRMSE as the following:

$$
\begin{aligned}
\mathrm{rMAE} &= \frac{\sum_{n=1}^{N} |\hat{u}(x_n, t_n) - u(x_n, t_n)|}{\sum_{n=1}^{N_{res}} |u(x_n, t_n)|} \\
\mathrm{rRMSE} &= \sqrt{\frac{\sum_{n=1}^{N} |\hat{u}(x_n, t_n) - u(x_n, t_n)|^2}{\sum_{n=1}^{N} |u(x_n, t_n)|^2}}
\end{aligned}
\tag{7}
$$

where $N$ is the number of testing points, $\hat{u}$ is the neural network approximation, and $u$ is the ground truth.

## B    APPENDIX B: PDEs SETUPS

We provide detailed PDE setups for convection, reaction-diffusion, and 1D-reaction equations.

**Convection PDE.** The one-dimensional convection problem is a hyperbolic PDE that is commonly used to model transport phenomena. The system has the formulation with periodic boundary conditions as follows:

$$\frac{\partial u}{\partial t} + \beta \frac{\partial u}{\partial x} = 0, \ \ \forall x \in [0, 2\pi], \ t \in [0, 1]$$
$$\texttt{IC:} u(x, 0) = \sin(x), \ \ \texttt{BC:} u(0, t) = u(2\pi, t) \tag{8}$$

where $\beta$ is the convection coefficient. As $\beta$ increases, the frequency of its solution goes higher, and it becomes harder for PINNs to approximate. Here, we set $\beta = 50$.

**1D-Reaction PDE.** The one-dimensional reaction problem is a hyperbolic PDE that is commonly used to model chemical reactions. The system has the formulation with periodic boundary conditions as follows:

$$\frac{\partial u}{\partial t} - \rho u(1 - u) = 0, \ \ \forall x \in [0, 2\pi], \ t \in [0, 1]$$
$$\texttt{IC:} u(x, 0) = \exp(-\frac{(x - \pi)^2}{2(\pi/4)^2}), \ \ \texttt{BC:} u(0, t) = u(2\pi, t) \tag{9}$$

where $\rho$ is the reaction coefficient. Here, we set $\rho = 5$. The equation has a simple analytical solution:

$$u_{\texttt{analytical}} = \frac{h(x) \exp(\rho t)}{h(x) \exp(\rho t) + 1 - h(x)} \tag{10}$$

where $h(x)$ is the function of the initial condition.

**1D-Wave PDE.** The 1D-Wave equation is a hyperbolic PDE that is used to describe the propagation of waves in one spatial dimension. It is often used in physics and engineering to model various wave phenomena, such as sound waves, seismic waves, and electromagnetic waves. The system has the formulation with periodic boundary conditions as follows:

$$\frac{\partial^2 u}{\partial t^2} - \beta \frac{\partial^2 u}{\partial x^2} = 0 \ \ \forall x \in [0, 1], \ t \in [0, 1]$$
$$\texttt{IC:} u(x, 0) = \sin(\pi x) + \frac{1}{2} \sin(\beta \pi x), \ \ \frac{\partial u(x, 0)}{\partial t} = 0 \tag{11}$$
$$\texttt{BC:} u(0, t) = u(1, t) = 0$$

where $\beta$ is the wave speed. Here, we are specifying $\beta = 3$. The equation has a simple analytical solution:

$$u(x, t) = \sin(\pi x) \cos(2\pi t) + \frac{1}{2} \sin(\beta \pi x) \cos(2\beta \pi t) \tag{12}$$

**2D Navier-Stokes PDE.** The 2D Navier-Stokes equation is a parabolic PDE that consists of a pair of partial differential equations that describe the behavior of incompressible fluid flow in two-dimensional space. They are widely used in fluid dynamics to model the motion of fluids, such as air and water, in various engineering and scientific applications. The system has the formulation as follows:

$$\frac{\partial u}{\partial t} + \lambda_1 (u \frac{\partial u}{\partial x} + v \frac{\partial u}{\partial y}) = -\frac{\partial p}{\partial x} + \lambda_2 (\frac{\partial^2 u}{\partial x^2} + \frac{\partial^2 u}{\partial v^2})$$
$$\frac{\partial v}{\partial t} + \lambda_1 (u \frac{\partial v}{\partial x} + v \frac{\partial v}{\partial y}) = -\frac{\partial p}{\partial y} + \lambda_2 (\frac{\partial^2 u}{\partial x^2} + \frac{\partial^2 u}{\partial v^2}) \tag{13}$$

where $u(t, x, y)$ and $v(t, x, y)$ are the $x$-component and $y$-component of the velocity field separately, and $p(t, x, y)$ is the pressure. Here, we set $\lambda_1 = 1$ and $\lambda_2 = 0.01$. The system does not have an explicit analytical solution, while the simulated solution is given by Raissi et al. (2019).

## C APPENDIX C: ADDITIONAL RESULTS

**Ablation Study on Activation Functions.** To investigate the effectiveness of the Wavelet activation function in PINNsFormer, we compare the performance differences using Wavelet than

ReLU, Sigmoid, and Sin activation functions over convection and 1D-reaction problems. In particular, we study the effects of using the same activation function in both the feed-forward layer and encoder/decoder layer (marked as ReLU, etc.) and changing the activation function of the encoder/decoder layer to LayerNorm (as vanilla Transformer does, marked as ReLU+LN, etc.). The evaluation results are shown in Table 6.

| Activation | Convection | | | 1D-Reaction | | |
|---|---|---|---|---|---|---|
| | Loss | rMAE | rRMSE | Loss | rMAE | rRMSE |
| ReLU | 0.5256 | 1.001 | 1.001 | 0.2083 | 0.994 | 0.996 |
| Sigmoid | 0.1618 | 1.112 | 1.223 | 0.1998 | 0.991 | 0.993 |
| Sin | 0.3159 | 1.074 | 1.141 | 4.9e-6 | 0.017 | 0.032 |
| ReLU+LN | 0.7818 | 1.001 | 1.002 | 0.2028 | 0.992 | 0.993 |
| Sigmoid+LN | 0.0549 | 0.941 | 0.967 | 0.2063 | 0.992 | 0.990 |
| Sin+LN | 0.3219 | 1.083 | 1.156 | 4.7e-6 | 0.016 | 0.033 |
| Wavelet | **3.7e-5** | **0.023** | **0.027** | **3.0e-6** | **0.015** | **0.030** |
| Wavelet+LN | NaN | NaN | NaN | 3.9e-6 | 0.018 | 0.037 |

Table 6: Results for solving convection and 1D-reaction equations using Transformer architecture with different activation functions. PINNsFormer (with Wavelet activation) consistently outperforms all other activation functions in terms of training loss, rMAE, and rRMSE

The ablation study results show two major conclusions: First, using wavelet activation shows constantly better performance than ReLU, Sigmoid, and Sin activations. In particular, Sin activation may show effectiveness in only certain cases, while Wavelet can generalize all cases well. Second, Introducing LayerNorm activation to the encoder/decoder does not significantly contribute to performance improvement. In contrast, LayerNorm activation may cause convergence issues when coupling with the Wavelet activation function for certain situations.

**Hyperparameter Sensitivity Study.** To investigate the possible difficulties in picking hyperparameters $k$ and $\Delta$, we compared the performance differences with a mesh choice of these two hyperparameters over the 1d-reaction problem. The evaluation results (relative-$\ell_2$ error, with failure modes bolded) are shown in Table 7.

| $\Delta t$ | k=3 | k=5 | k=7 | k=10 |
|---|---|---|---|---|
| 1e-1 | 0.044 | **0.514** | **0.743** | **0.731** |
| 1e-2 | 0.029 | 0.035 | 0.045 | 0.049 |
| 1e-3 | 0.037 | 0.037 | 0.024 | 0.035 |
| 1e-4 | **0.997** | 0.030 | 0.029 | 0.046 |
| 1e-5 | **0.977** | 0.026 | **0.977** | 0.021 |

Table 7: Results for solving 1D-reaction equation with various combinations of $\Delta t$ and $k$. PINNsFormer shows the flexibility of a wide choice of hyperparameters on certain problems.

The study on hyperparameter sensitivity of $\Delta t$ and $k$ exhibits three intuitions: First, given a mesh choice of $k$ and $\Delta t$, PINNsFormer is not sensitive to a wide range of the two hyperparameters. For instance, PINNsFormer successfully mitigates the failure modes for any combinations of $k \in [1e-2, 1e-3, 1e-4]$ and $\Delta t \in [3, 5, 7]$. Second, the choice of $\Delta t$ should not be either too large (i.e., 1e-1) or too small (i.e., 1e-5). Intuitively, either a too-large or a too-small $\Delta t$ degrades the temporal dependencies between discrete time steps. Third, increasing the pseudo-sequence length can help mitigate PINNs failure modes (i.e., $k = 3 \to 5$ when $\Delta t = 1e-4$). However, once PINNs successfully mitigate the failure mode, the benefit of further increasing $k$ is marginal.

**Result Visualizations.** We here present the plots of ground truth solutions, neural network predictions, and absolute errors for all evaluations included in the experimental section. The plots on convection, 1D-reaction, 1D-wave, and 2D Navier-Stokes equations are shown in Figure separately.

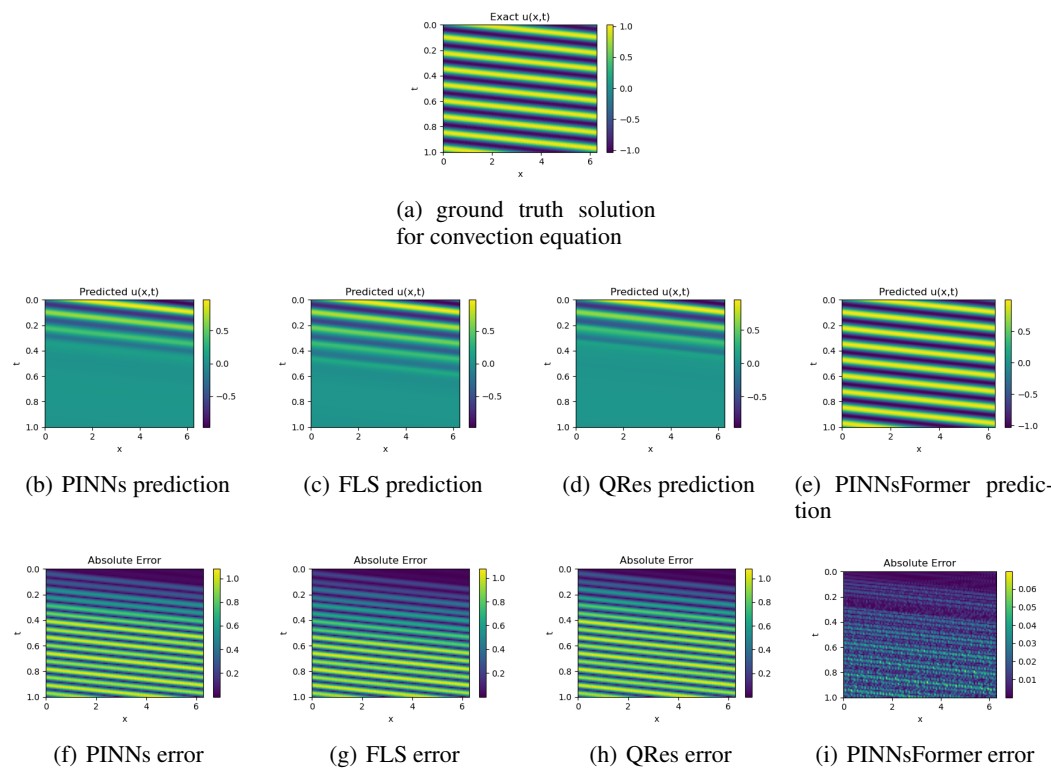

Figure 6: Ground truth solution, predictions, and absolute errors (up to bottom) of PINNs, FLS, QRes, PINNsFormer (left to right) over convection equation.

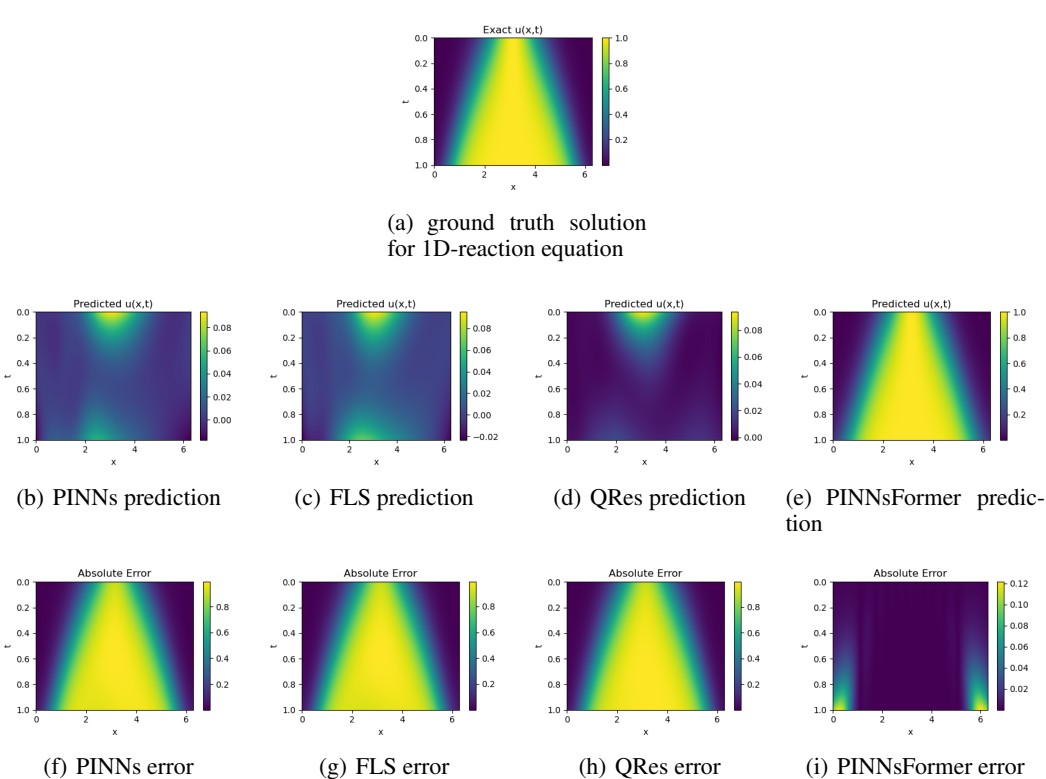

Figure 7: Ground truth solution, predictions, and absolute errors (up to bottom) of PINNs, FLS, QRes, PINNsFormer (left to right) over 1D-reaction equation.

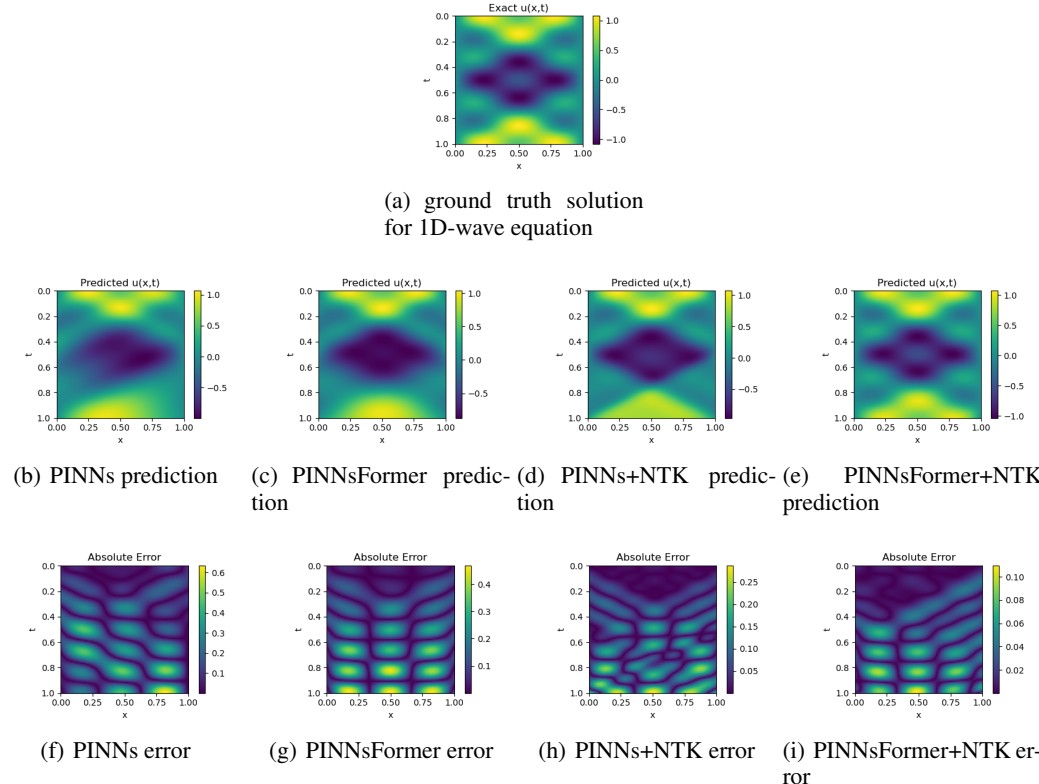

(a) ground truth solution
for 1D-wave equation

(b) PINNs prediction  (c) PINNsFormer predic-  (d) PINNs+NTK predic-  (e) PINNsFormer+NTK
tion                     tion                     prediction

(f) PINNs error  (g) PINNsFormer error  (h) PINNs+NTK error  (i) PINNsFormer+NTK er-
ror

Figure 8: Ground truth solution, predictions, and absolute errors (up to bottom) of PINNs, PINNs-Former, PINNs+NTK, PINNsFormer+NTK (left to right) over 1D-reaction equation.

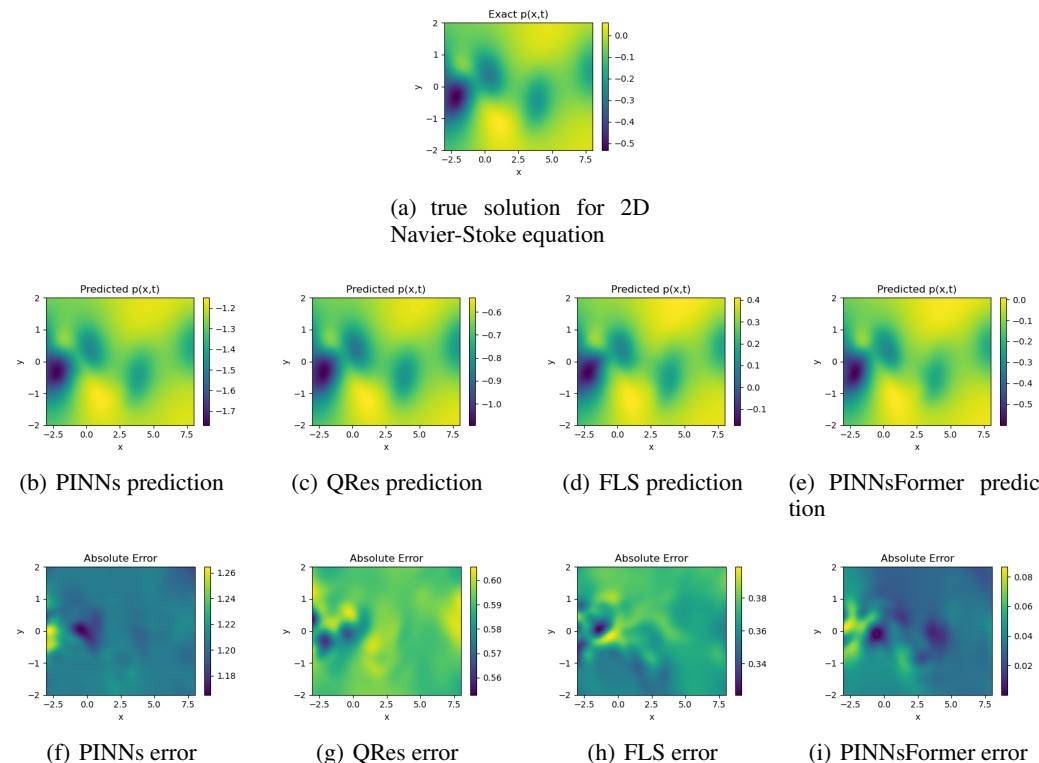

(a) true solution for 2D
Navier-Stoke equation

(b) PINNs prediction  (c) QRes prediction  (d) FLS prediction  (e) PINNsFormer predic-
tion

(f) PINNs error  (g) QRes error  (h) FLS error  (i) PINNsFormer error

Figure 9: Ground truth solution, predictions, and absolute errors (up to bottom) of PINNs and PINNsFormer (left to right) over 2D Navier-Stokes equation.

