# OpenReview forum: "PINNsFormer: A Transformer-Based Framework For Physics-Informed Neural Networks"
_ICLR.cc/2024/Conference — ICLR 2024 poster_

### Official Review · Reviewer_dren · 2023-10-26

**Soundness:** 2 fair
**Presentation:** 2 fair
**Contribution:** 1 poor
**Rating:** 5
**Confidence:** 3

**Summary:**

This paper presents a method to utilize the well-known transformer architecture in PINNs. Unlike PINN’s point-to-point processing, the proposed method produces multiple outputs in a forward pass by injecting multiple input coordinates. The authors used a ‘pseudo-sequence generator’, which constructs a sequence of input coordinates consisting of a spatial coordinate and multiple time coordinates. The following transformer module models the dependency between input coordinates to generate the final outputs. They also introduce wavelet activation function that shows the effectiveness. The authors have tested on three different PDEs, and it shows the comparative performance.

**Strengths:**

1. Using transformers in PINN training is a promising research area. I appreciate the authors’ attempt to incorporate it into PINNs.
2. The idea of processing multiple coordinates seems interesting and original.

**Weaknesses:**

1. I respectfully disagree with the author’s argument that the original PINN neglects temporal dependency. PINN takes temporal coordinates and spatial coordinates together and generates the output. By going through multiple layers in MLP, the time coordinate will definitely affect the features from the spatial coordinates. I agree that the suggested method might be able to model time dependency more explicitly. However, the argument that the original PINN is not capable of modeling time dependency is too strong.

2. As a follow-up comment, we sample many random collocation points at each iteration, and MLP can see many time coordinates with spatial coordinates during training. Hence, I believe the original PINN is capable of modeling time dependency.

3. Is delta t fixed? if yes, then it might be a not trivial limitation, considering different time granularity at different time coordinates.

4. If I understand correctly, it seems that Spatio-Temporal Mixer is just one layer MLP. I might have missed something, but the technical details are not properly described in explaining each component. The formal definition of each module would be appreciated.

5. The proposed Wavelet activation function seems to be a simplified version of the positional encoding. In addition, IMHO, the sine activation function, followed by MLP, which attaches weights to each neuron, could do the same thing. cos function can also be easily expressed by a bias term from the previous layer e.g., sin(x) = cos(90-x).

6. My main concern is a weak experimental setup and results. The authors presented the results of simple three PDEs, which already have been tackled by numerous works. And, there are many previous works that achieved better results (lower relative errors) [1].
The authors highlighted ‘very low loss’ values. Loss depends on loss function and hyperparameters, which cannot be a fair metric.
The Navier-stokes experiment used in this paper is too simple. Please consider using examples tested in [2] and [3].

[1] An expert guide’s to training physics-informed neural networks, Wang et al., arXiv 2023.

[2] Respecting causality is all you need for training physics-informed neural networks, Wang et al., arXiv 2022.

[3] Separable physics-informed neural networks, Cho et al., NeurIPS 2023.

**Questions:**

Questions are embedded in the section above.

---

> ### Author Response · Authors · 2023-11-15
> **Rebuttal to Reviewer dren**
>
> We thank the reviewer for the thoughtful comments on our paper.
>
> **Response to W1 & W2:**
>
> We respectfully disagree with the reviewer’s statements regarding capturing temporal dependencies with MLP-based PINNs. While it's true that MLP-based PINNs incorporate both spatial and temporal inputs, MLP-based PINNs process different inputs (i.e., batch data) independently. For standard MLP architectures, each spatio-temporal input (even with random samplings of collocation points) is treated as a distinct entity, and the model processes them separately without explicitly capturing the temporal dependencies between consecutive time steps. In contrast, PINNsFormer explicitly suggests capturing temporal dependencies within local consecutive time steps when processing each spatiotemporal input. In our experiment section, we highlight the function of PINNsFormer in explicitly capturing temporal dependencies rather than MLP-based PINNs, with clear performance benefits by empirical evaluations.
>
> **Response to W3:**
>
> No, $\Delta t$ is not fixed, it is a tunable hyperparameter. We have provided an additional ablation study regarding the different choices of $\Delta t$ in the global common response section.
>
> **Response to W4:**
>
> Yes, Spatio-Temporal Mixer in our implementation is just a one-layer MLP. The Spatio-Temporal Mixer module serves as an embedding process as in the vanilla Transformer. We show that a one-layer MLP is enough and effective to embed low-dimensional spatiotemporal information in high-dimensional spaces based on empirical evaluations.
>
> **Response to W5:**
>
> We respectfully disagree with the reviewer’s statement.
>
> First, Positional Encoding (PE) is a typical technique used in combination with embedding methods, it serves different purposes as activation functions, including Wavelet. Also, PE and Wavelet have different mathematical expressions.
>
> Second, Wavelet activation is not the same as Sin activation. We assign two learnable parameters $\omega_1$ and $\omega_2$ to weigh the sin and cos components in Wavelet activation. Indeed, Wavelet can be simplified to $k \sin (x+ \phi)$ for some $k$ and $\phi$ (note $\phi$ does not necessarily need equal to $\pi/2$). However, on the one hand, intuitively, compared to pure Sin activation, Wavelet introduces additional phase and magnitude information. The arbitrary learnable combination of phase and magnitude gives a larger capability than the pure Sin activation function with fixed phase and magnitude information. On the other hand, precisely, let assume $\sin (x_1) = k_1 \sin(x_2)+ k_2 \cos(x_2)$, where $x_1$ and $x_2$ are the arbitrary outputs from two independent linear layers. To reach the equality, we need have $x_2 = \arcsin(\frac{k_1}{\sqrt{k_1^2+k_2^2}}) + \arccos(\frac{\sin x}{\sqrt{k_1^2+k_2^2}})$, the translation from $x_1$ to $x_2$ involves non-linear operations, which cannot be achieved by simply add additional weights/bais to neurons. Our ablation study 1 in the global common response also empirically demonstrates the difference between Sin and Wavelet activation function.
>
> **Response to W6:**
>
> We respectfully disagree with the reviewer’s statement.
>
> First, all our evaluations strictly follow existing prior works (convection and 1d-diffusion as [1], 1d-wave as [2], and Navier-Stokes as [3]). By evaluating these problems, we can make clear comparisons with existing baselines.
>
> Second, we use MSE loss for all evaluations, which is widely used in PINNs. Comparing all evaluations under the same training loss metric does not lead to unfair results. If the reviewer meant for the impact of model hyperparameters, we fine-tune each model, while additionally controlling for a roughly similar number of model parameters for all baselines and PINNsFormer to avoid the effect of overparameterizations.
>
> Third, our Navier-Stokes equation is essentially the same as the reference papers. We are using the fundamental formulation of the Navier-Stokes equation while they are using the velocity-vorticity formulation. Both formulations are mathematically equivalent, and thus, we do not believe the NS equation in our setting is simpler than the reference papers. Their showcases seem to be more complicated than ours because they are visualizing $u, v, \omega$ and we are visualizing $p$.
>
> [1] Krishnapriyan, Aditi, et al. "Characterizing possible failure modes in physics-informed neural networks." Advances in Neural Information Processing Systems 34 (2021): 26548-26560.
>
> [2] Wang, Sifan, Xinling Yu, and Paris Perdikaris. "When and why PINNs fail to train: A neural tangent kernel perspective." Journal of Computational Physics 449 (2022): 110768.
>
> [3] Raissi, Maziar, Paris Perdikaris, and George E. Karniadakis. "Physics-informed neural networks: A deep learning framework for solving forward and inverse problems involving nonlinear partial differential equations." Journal of Computational physics 378 (2019): 686-707.

---

> > ### Author Response · Authors · 2023-11-21
> > **Response to Reviewer dren**
> >
> > We hope this message finds you well.
> >
> > As the deadline for final evaluations is tomorrow, we would greatly appreciate any insights or suggestions you might have to our previous response, which includes two additional ablation studies and additional clarifications on you previous concerns. We would also be grateful for any consideration towards improving our current score based on potential revisions.
> >
> > Thank you very much for your time and understanding.

---

> > > ### Comment · Reviewer_dren · 2023-11-22
> > > **Response**
> > >
> > > W1 & W2
> > > -> Thanks for your response, and I agree with your argument, though I still think that the argument "PINNs neglect the crucial temporal dependencies" is too strong. You may argue that the suggested method is 'explicitly' handling temporal dependency, but I think the claim that PINNs are incapable of modeling temporal dependency is not true. It's trained w/ SGD, given numerous time and spatial coordinates as a batch, and it will minimize PDE residuals for all coordinates together.
> > >
> > > W3
> > > -> Thanks for the clarification. It's a hyperparameter, meaning it's fixed during training. It would be nicer if there exists a solution for handling varying \delta during training, for example, different \delta t during every iteration during training, but you can add some positional encoding to differentiate time intervals.
> > >
> > > W4 & W5
> > > -> I still think the experimental results are weak. For Convection and reaction equations, if we use recent techniques, we can easily get lower relative errors than the results in the proposed methods.
> > > -> You presented 'Low', 'rMAE', 'rRMSE', I did not complain about 'rMAE' and 'rRMSE', I was curious about 'Loss', each coefficient for each loss (initial condition loss, residual loss, boundary loss, etc.) makes fair comparison hard. Also, MSE loss (not relative L2 loss) may exaggerate the performance improvement.
> > > -> NS equation, I mentioned about more turbulent and complex initial conditions. The experimental setting you presented can be easily solved by conventional PINNs w/ recent training techniques.
> > >
> > > Overall, the authors answered many questions, and I do appreciate it. I will increase my score to 5.

---

### Official Review · Reviewer_mmgn · 2023-10-30

**Soundness:** 3 good
**Presentation:** 3 good
**Contribution:** 3 good
**Rating:** 5
**Confidence:** 4

**Summary:**

This paper introduces PINNsFormer, a novel transformer-based framework for Physics-Informed Neural Networks (PINNs) to approximate solutions to partial differential equations (PDEs). PINNsFormer addresses the limitation of conventional PINNs in neglecting temporal dependencies within PDEs. Comprehensive experiments show PINNsFormer outperforms PINNs and variants in addressing failure modes and high-dimensional PDEs.

**Strengths:**

PINNsFormer addresses a key limitation of PINNs by explicitly learning temporal dependencies, crucial for real-world physics systems. This significantly improves PINNs' generalization ability.

The proposed pseudo sequence representation and transformer architecture are clever approaches to adapt PINNs for sequential models.

Ablation studies provide insights into design choices and integration of existing PINNs schemes.

**Weaknesses:**

While the Wavelet activation function is theoretically justified to approximate arbitrary solutions, its advantages over other activations like ReLU, sigmoid, etc. require further empirical analysis and validation on practical problems. Conducting detailed empirical studies to evaluate Wavelet against various state-of-the-art activations under different settings can provide better insights into its benefits and limitations. This is important to fully understand its behavior and assess its effectiveness.

The paper only considers isotropic problems which have constant properties in all directions. However, most real-world physics systems exhibit anisotropic and nonlinear characteristics. Extending PINNsFormer to handle anisotropic problems modeled by direction-dependent PDEs, as well as nonlinear problems involving variable coefficients, would significantly broaden its applicability and demonstrate the approach's versatility.

No quantitative analysis was performed to evaluate important design choices like the pseudo sequence length and number of levels in coarsening. Without such ablation studies, it is difficult to justify critical hyperparameters and understand their impact on the model's performance as well as computational efficiency. These quantitative studies would provide further insights to validate the architectural design of PINNsFormer.

Although various benchmark problems were tested, stronger validation would involve demonstrating the approach's effectiveness in entirely new physical domains beyond the existing test cases. Without such generalization to unseen problem classes, the claims regarding PINNsFormer's broad applicability remain partially unsubstantiated.

While efficient on smaller problems, the inherent quadratic complexity of self-attention may pose scalability challenges for extremely large spatiotemporal datasets. Developing techniques to alleviate this computational limitation would enhance the method's practicality when dealing with massive real-world physics simulations.

**Questions:**

see weakness above

---

> ### Author Response · Authors · 2023-11-15
> **Rebuttal to Reviewer mmgn**
>
> We thank the reviewer for the thoughtful comments on our paper.
>
> **Response to ‘Missing ablation studies on different activation functions’:**
>
> We are happy to add the ablation study on using different activation functions. We include the ablation setups, results, and conclusions in the global common response. Overall, our additional ablation shows that using Wavelet activation shows constantly better performance than ReLU, Sigmoid, and Sin activations. In particular, Sin activation shows effectiveness only in certain cases, while Wavelet can generalize all cases well.
>
> **Response to ‘Considers isotropic problems only’:**
>
> First, we strictly follow the exact same setups of PINNs as most prior works do [1,2,3], and our explorations based on existing setups and our works are meaningful. Second, the main purpose of this paper is to bring the temporal dependency idea to PINNs, with a natural adaptation of transformer architectures. Investigating anisotropic and nonlinear characteristics in physics systems with PINNsFormer is worth future explorations but yet out of our scope. For instance, a natural idea to handle direction-dependent PDEs is to introduce patches (as ViT [4] does) rather than pseudo sequences, which allows capturing higher-order dependencies than simple temporal dependencies.
>
> **Response to ‘Missing ablation study on pseudo-sequence generator’:**
>
> We are happy to add the ablation study on hyperparameters of the pseudo-sequence generator. We include the ablation setups, results, and conclusions in the global response. Please see the response above. Overall, given a mesh choice of $k$ and $\Delta t$, PINNsFormer is not sensitive to a wide range of $k$ and $\Delta t$. PINNsFormer successfully mitigates the failure modes for any combinations of $k\in [1e-2, 1e-3, 1e-4]$ and $\Delta t \in [5,7,10]$. However, we observe that either a too-large or a too-small $\Delta t$ may degrade the temporal dependencies between each time step. In the last, Increasing the pseudo-sequence length can help mitigate PINNs failure modes, but it only provides marginal benefit once our models have mitigated the failure mode/
>
>
> **Response to ‘Extrapolation validation should be included’:**
>
> We agree that understanding and mitigating the extrapolation (training and testing data are from different domains) failures of PINNs is a novel and challenging problem [5]. However, our paper focuses on interpolation validation as most prior PINNs works do, with showing clear benefits using PINNsFormer. The extrapolation ability of PINNsFormer is worth future explorations (maybe by adapting PINNsFormer to a forecasting model), but it is either out of our scope or not the main purpose of most PINNs works.
>
> **Response to ‘Improving the quadratic complexity’:**
>
> We agree that involving full attention will cause a $\mathcal{O}(L^2)$ complexity. However, there are existing methods such as replacing full-attention with prob-sparse self-attention [6], which reduces to $\mathcal{O}(L\log L)$ complexity. Our work aims to bring explicit temporal dependencies into PINNs, with an adaptation to Transformer as a natural solution. Improving the quadratic complexity in Transformers such as [6] is complementary future work but is currently out of our scope.
>
> [1] Krishnapriyan, Aditi, et al. "Characterizing possible failure modes in physics-informed neural networks." Advances in Neural Information Processing Systems 34 (2021): 26548-26560.
>
> [2] Wang, Sifan, Xinling Yu, and Paris Perdikaris. "When and why PINNs fail to train: A neural tangent kernel perspective." Journal of Computational Physics 449 (2022): 110768.
>
> [3] Raissi, Maziar, Paris Perdikaris, and George E. Karniadakis. "Physics-informed neural networks: A deep learning framework for solving forward and inverse problems involving nonlinear partial differential equations." Journal of Computational physics 378 (2019): 686-707.
>
> [4] Dosovitskiy, Alexey, et al. "An image is worth 16x16 words: Transformers for image recognition at scale." arXiv preprint arXiv:2010.11929 (2020).
>
> [5] Fesser, Lukas, Richard Qiu, and Luca D'Amico-Wong. "Understanding and Mitigating Extrapolation Failures in Physics-Informed Neural Networks." arXiv preprint arXiv:2306.09478 (2023).
>
> [6] Zhou, Haoyi, et al. "Informer: Beyond efficient transformer for long sequence time-series forecasting." Proceedings of the AAAI conference on artificial intelligence. Vol. 35. No. 12. 2021.

---

> ### Author Response · Authors · 2023-11-21
> **Response to Reviewer mmgn**
>
> We hope this message finds you well.
>
> As the deadline for final evaluations is tomorrow, we would greatly appreciate any insights or suggestions you might have to our previous response, which includes two additional ablation studies and additional clarifications on you previous concerns. We would also be grateful for any consideration towards improving our current score based on potential revisions.
>
> Thank you very much for your time and understanding.

---

### Official Review · Reviewer_ZyRg · 2023-10-31

**Soundness:** 4 excellent
**Presentation:** 4 excellent
**Contribution:** 4 excellent
**Rating:** 8
**Confidence:** 5

**Summary:**

The manuscript describes a novel architecture for PINNs where a sequence with  even time steps is created.. This forms a sequence that can be used as an input to a transformer. The first layer of the neural network has a special "wavelet" non-linearity that can be seen as a spectral type of embedding of the data, with trainable parameters for the amplitudes and the frequency. An additional projection is made before the encoder and decoder. and the final output is generated with fully connected layer.

An attention mechanism is trained to produce the function value at   $ \hat u( x , t_i + k \Delta t )$ from the values   $ \\{ \\hat u( x , t_i + j \Delta t ) \\}_{j\in \\{1,..,k-1 \\}} $

The described architecture is tested against a set of baselines with impressive results.

**Strengths:**

Impressive results.  Clear representation.

**Weaknesses:**

The system is using a time discretised set of function values to predict the next time step. This reminds me of the finite difference method, In this case the stencil is 100 elements long, so the optimum stencil would have a very high order in accuracy.

The reason why very large stencils are not used is that these bear a computational cost, and the same happens in using attention, although most of the computing is parallel.

Now the manuscript does not provide a baseline using a normal, discrete PDE solver, of course,  on the par with the  computational load that the attention mechanism is requiring . Of course, having an analytical, although complicated, solution for a problem has its advantages compared to a set of discrete nodal values.

I would like that the authors would address this in their submission for better rating.

**Questions:**

See the weakness part.

---

> ### Author Response · Authors · 2023-11-15
> **Rebuttal to Reviewer ZyRg**
>
> We thank the reviewer's recognition of our work. We want to argue three points to the statements in Weakness.
>
> **(1) Similarities and Differences with Finite Difference Method:**
>
> Despite the sequential input in PINNsFormer, which can be similar to the finite difference method, they are actually different. The key idea of the finite difference method is that it discretizes the whole domain and propagates the solution globally and sequentially. PINNsFormer, instead, focuses on the temporal dependency by looking at additional collocation points near the original collocation point. In PINNsFormer, the predictions within a sequence do not have a specific sequential propagation order, but all predictions utilize temporal dependencies from each other for better predictions in parallel.
>
> **(2) Computational Cost:**
>
> Using attention mechanisms or complicated architectures can indeed raise additional computational costs. However, there are existing solutions, such as Informer [1], which uses prob-sparse attention to reduce the computation cost from $\mathcal{O} (L^2)$ to $\mathcal{O}(L\log L)$. However, the main purpose of this paper is to bring the temporal dependency idea to PINNs with a natural adaptation of transformer architectures. Investigating such mechanisms in PINNsFormer is worth future explorations but yet out of our scope.
>
> **(3) Adding Normal, Discrete PDE Solver as Baseline:**
>
> Here, we recognize the normal, discrete PDE solver as the finite element method (FEM) as the reviewer mentioned. However, FEM is typically not a baseline in the context of PINNs. Instead, research works on PINNs [2] use simulation results from FEM as the ground truth to evaluate the neural networks’ performance. Hence as such, we did not include FEM as a baseline when comparing to PINNs and their variations.
>
> [1] Zhou, Haoyi, et al. "Informer: Beyond efficient transformer for long sequence time-series forecasting." Proceedings of the AAAI conference on artificial intelligence. Vol. 35. No. 12. 2021.
>
> [2] Raissi, Maziar, Paris Perdikaris, and George E. Karniadakis. "Physics-informed neural networks: A deep learning framework for solving forward and inverse problems involving nonlinear partial differential equations." Journal of Computational physics 378 (2019): 686-707.

---

> > ### Author Response · Authors · 2023-11-21
> > **Response to Reviewer  ZyRg**
> >
> > We hope this message finds you well.
> >
> > As the deadline for final evaluations is tomorrow, we would greatly appreciate any insights or suggestions you might have to our previous response, which includes two additional ablation studies and additional clarifications on you previous concerns. We would also be grateful for any consideration towards improving our current score based on potential revisions.
> >
> > Thank you very much for your time and understanding.

---

> > > ### Comment · Reviewer_ZyRg · 2023-11-22
> > >
> > > Thank you for your clarifications.
> > >
> > > I think this is a good manuscript and should be published ICLR. My reflection on your comments:
> > >
> > >  (1)   Let me be clearer what on what I meant. The prediction you are using looks like an end result of this recursive finite difference expansion.
> > > $$ f(t_0 + Nt) \approx f(t_ 0 + (N-1)t) + t  f'(t_ 0 + (N-1)t)$$
> > > or
> > > $$f(t_ 0 + Nt) \approx 2f(t_0 + (N-1)t) - f(t_ 0 + (N-2)t) + t^2 \cdot \
> > > f''(t_ 0 + (N-1)t)$$
> > > etc.
> > >
> > > By using higher and higher order stencils one can continue this even to f(t_0) always increasing the order. Of course one could use the derivatives as well as they are also available. This is definitely a use of finite differences.  The larger the stencil is, the more orders one has the more accurate the approximation is.  My assumption is that the trained attention model will reproduce the coefficients of this expression for the prediction of the last temporal value.
> > > The spatial dimensions do not matter here, but using the PDE on could express the time derivatives with spatial derivatives.....
> > >
> > > (2&3)  In my option about reviewing papers has two major goals; it should increase understanding, but it is even better if it would provide valuable practical means to solve relevant problems.  That is why I asked about comparison to FEM as a baseline, as the PINNS are practical only if they provide more accurate solutions and solve them faster in real time. I admit that comparison to GPU accelerated parallelised FEM codes (to be on equal ground with PINNs) amounts to quite a lot of work.

---

### Official Review · Reviewer_PNtx · 2023-11-07

**Soundness:** 3 good
**Presentation:** 4 excellent
**Contribution:** 3 good
**Rating:** 8
**Confidence:** 4

**Summary:**

This paper proposes PINNsFormer, a novel Transformer-based framework for Physics-Informed Neural Networks (PINNs). PINNs are used to numerically solve partial differential equations (PDEs) but struggle to capture temporal dependencies inherent in physics systems. PINNsFormer addresses this by generating pseudo input sequences from pointwise inputs and using a Transformer encoder-decoder architecture to model temporal dependencies. The main contributions are: 1) A new framework called PINNsFormer that equips PINNs with the ability to capture temporal dependencies through generated pseudo sequences and Transformer architecture. 2) A novel activation function called Wavelet designed to anticipate Fourier decomposition. 3) Extensive experiments showing PINNsFormer outperforms PINNs on problems involving failure modes and high-dimensional PDEs. 4) Demonstration of flexibility to incorporate PINNs optimization schemes for enhanced performance.

**Strengths:**

1. **Novelty**: This is the first work I'm aware of that integrates Transformers with PINNs to capture temporal dependencies,  which is a novel and promising direction. Adapting Transformers designed for sequences to point-wise PINNs is non-trivial, thus the innovations in pseudo-sequence generation and loss formulation are important contributions.

2. **Contributions**: The results on problems like convection and 1D-reaction demonstrate clear benefits in preventing temporal propagation failures that cripple vanilla PINNs. In addition, this work shows that modeling inter-timestep dependencies appears highly effective in maintaining accuracy across the domain rather than just near initial conditions.

3. **Methodology**: The model components are well-motivated - the pseudo-sequence generation and Transformer encoding seem natural yet powerful ways to incorporate temporal modeling into PINNs.

4. **Writing**: The paper is very clearly written, laying out both the background and proposed methodology comprehensively.

**Weaknesses:**

1. The lack of published code or detailed hyperparameters makes reproducibility difficult. Providing an implementation would strengthen the paper's contributions.

2. While the overall approach is promising, some ablation studies would help determine the impact of different components like the pseudo-sequence generation and Wavelet activation.

3. Since PINNs are notoriously slow in training, the computational overhead of PINNsFormer could be prohibitive for some large-scale applications. Analysis of model complexity and efficiency could help elucidate this issue.

4. The reliance on introducing a discrete timestep risks undermining the automatic differentiation advantage of PINNs. Justification for this design choice could be expanded.

**Questions:**

1. One major advantage of PINNs is that, it leverages automatic differentiation rather than relying on finite difference approximations. On the other hand, the use of discrete pseudo-sequences means temporal dependencies are modeled in the fashion of finite difference approximations rather than pure automatic differentiation. Would it create any difficulties in picking parameters for this differentiation (e.g., the Δt)?

A relevant thought: The impact of the timestep granularity Δt seems worth further analysis. Is there a study on model sensitivity to this parameter? Does the performance degrade if Δt is too small or large? Are there any guidelines for setting Δt?

2. It may be worth trying the causal-attention transformers (decoder-based, e.g. LLM) instead of the encoder-decoder architecture. It does not seem to have particular reasons to adopt current sequence-to-sequence architecture.

3. Other sampling schemes may also play an important role [1], especially for mitigating the temporal propagation failure in PINNs. Since PINNFormer relies on discretization on temporal dimensions, I am curious how PINNFormer can adapt to non-fixed sampling. I believe the transformer architectures have such flexibility.

[1] Daw, Arka & Bu, Jie & Wang, Sifan & Perdikaris, Paris & Karpatne, Anuj. (2022). Rethinking the Importance of Sampling in Physics-informed Neural Networks. 10.48550/arXiv.2207.02338.

---

> ### Author Response · Authors · 2023-11-15
> **Rebuttal to Reviewer PNtx**
>
> We thank the reviewer for the thoughtful comments for our paper.
>
> **Response to W1:**
>
> We have included the detailed implementation and hyperparameters in our paper. We include model hyperparameters in Table 4, Appendix A. We include all implementation, including demos for all experiments included in the paper (with fixed random seed for reproducibility purposes), through an anonymous GitHub repo: https://anonymous.4open.science/r/pinnsformer_iclr-4D09, as mentioned on page 7.
>
> **Response to W2:**
>
> Thanks. We are happy to provide additional ablation studies on (1) the effects of different hyperparameters in pseudo-sequence generators, and (2) the effects of using different activation functions. Please check the common response for details. Overall, our additional ablations show that (i) using Wavelet activation shows constantly better performance than ReLU, Sigmoid, and Sin activations, (ii) $k$ and $\Delta t$ show non-sensitivity to PINNsFormer's performance in general and can be chosen from a wide range.
>
> **Response to W3:**
>
> We include the computational and memory overheads in Table 5, Appendix A. Empirically, we demonstrate that the computation and memory overheads increase sublinearly as the pseudo-sequence length increases. In particular, using $k=5$ is already good enough for all experiments included, which yields tolerably approximately 3x computational cost and 2x memory cost. Theoretically, existing works such as informer [1] have been shown to reduce the complexity of attention from $\mathcal{O}(L^2)$ to $\mathcal{O}(L\log L)$ by replacing full-attention to prob-sparse attention. Investigating such mechanisms in PINNsFormer is worth future explorations but yet out of our scope.
>
> **Response to W4 & Q1:**
>
> We do use automatic differentiation rather than relying on finite difference approximations, so the advantage of PINNs is also inherited in PINNsFormer. In particular, despite a Transformer architecture to consider the temporal dependencies over multiple discrete time steps, the gradient calculation, as specified in Equation 5, is still executed point-wisely through automatic differentiation. We are happy to add more detailed explanations in the final version. The investigation of different $\Delta t$s is included in Response to W2.
>
> **Response to Q2:**
>
> We thank the reviewer’s suggestion on causal-attention transformers. However, the main purpose of this paper is to bring the temporal dependency idea to PINNs, with a natural adaptation of transformer architectures. Investigating more advanced transformer-based architectures is worth future explorations but yet out of our scope.
>
> **Response to Q3:**
>
> The adaptive sampling strategy can be easily incorporated into PINNsFormer, as we have demonstrated the flexibility of incorporating PINNsFormer with existing training schemes in Section 4.3. For instance, the mentioned paper uses adaptive samplings according to the computed risks. One can also incorporate this to PINNsFormer by (1) input collocation point (x,t), (2) output sequential output $[ \hat{u}(x,t+i\Delta t)]_{i=0}^{k-1}$, (3) compute the weight based on either the sum of the risk of all elements of the sequence or the risk of the first element of the sequence.
>
> [1] Zhou, Haoyi, et al. "Informer: Beyond efficient transformer for long sequence time-series forecasting." Proceedings of the AAAI conference on artificial intelligence. Vol. 35. No. 12. 2021.

---

> ### Author Response · Authors · 2023-11-21
> **Response to Reviewer PNtx**
>
> We hope this message finds you well.
>
> As the deadline for final evaluations is tomorrow, we would greatly appreciate any insights or suggestions you might have to our previous response, which includes two additional ablation studies and additional clarifications on you previous concerns. We would also be grateful for any consideration towards improving our current score based on potential revisions.
>
> Thank you very much for your time and understanding.

---

> > ### Comment · Reviewer_PNtx · 2023-11-23
> >
> > I appreciate the comments from the authors. Since I have already acknowledged the contributions of the paper, I will keep the score as it is.

---

### Author Response · Authors · 2023-11-15
**Rebuttal to All Reviewers (Common Response)**

As asked by several reviewers regarding the ablation study, we are happy to add the following two experiments:

**Ablation Study 1: Effect of Different Activation Functions:**

To investigate the effectiveness of the Wavelet activation function in PINNsFormer, we compare the performance differences using Wavelet than ReLU, Sigmoid, and Sin activation functions over convection and 1D-reaction problems. In particular, we study the effects of using the same activation function in both the feed-forward layer and encoder/decoder layer (marked as ReLU, etc.), and changing the activation function of the encoder/decoder layer to LayerNorm (as vanilla Transformer does, marked as ReLU+LN, etc.). The evaluation results are shown in the following table.


|   Convection   | Training Loss | r-$l_1$ error | r-$l_2$ error |  1D-Reaction   | Training Loss | r-$l_1$ error | r-$l_2$ error |
|:--------------:|:-------------:|:-------------:|:-------------:|:--------------:|:-------------:|:-------------:|:-------------:|
| ReLU       | 0.5256        | 1.001         | 1.001         | ReLU        | 0.2083        | 0.994         | 0.996         |
| Sigmoid    | 0.1618        | 1.112         | 1.223         | Sigmoid     | 0.1998        | 0.991         | 0.993         |
| Sin        | 0.3159        | 1.074         | 1.141         | Sin         | 4.9e-6        | 0.017         | 0.032         |
| ReLU+LN    | 0.7818        | 1.001         | 1.002         | ReLU+LN     | 0.2028        | 0.992         | 0.993         |
| Sigmoid+LN | 0.0549        | 0.941         | 0.967         | Sigmoid+LN  | 0.2063        | 0.992         | 0.990         |
| Sin+LN     | 0.3219        | 1.083         | 1.156         | Sin+LN      | 4.7e-6        | 0.016         | 0.033         |
| Wavelet    | **3.7e-5**        | **0.023**         | **0.027**         | Wavelet     | **3.0e-6**        | **0.015**         | **0.030**         |
| Wavelet+LN | NaN           | NaN           | NaN           | Wavelet+LN  | 3.9e-6        | 0.018         | 0.037         |

**Conclusions to Ablation Study 1:**

(1) Using wavelet activation shows constantly better performance than ReLU, Sigmoid, and Sin activations. In particular, Sin activation may show effectiveness in only certain cases, while Wavelet can generalize all cases well.

(2) Introducing LayerNorm activation to encoder/decoder does not significantly contribute to performance improvement. In contrast, LayerNorm activation may cause convergence issues when coupling with the Wavelet activation function for certain situations.

**Ablation Study 2: Effect on Hyperparameters of Pseudo-Sequence Generator:**

To investigate the possible difficulties in picking $k$ and $\Delta t$, we compared the performance differences with a mesh choice of these two hyperparameters over the 1d-reaction problem. The evaluation results (relative-$l_2$ error, with failure modes bolded) are shown in the following table.

| $\Delta t$ | k=3   | k=5   | k=7   | k=10  |
|:--------------:|:-------------:|:-------------:|:-------------:|:--------------:|
| 1e-1    | 0.044| **0.514**| **0.743**| **0.731**|
| 1e-2    | 0.029| 0.035| 0.045| 0.049|
| 1e-3    | 0.037| 0.037| 0.024| 0.035|
| 1e-4    | **0.997**| 0.030| 0.029| 0.046|
| 1e-5    | **0.977**| 0.026| **0.977**| 0.021|

**Conclusion to Ablation Study 2:**

(1) Given a mesh choice of $k$ and $\Delta t$, PINNsFormer is not sensitive to a wide range of $k$ and $\Delta t$. For instance, PINNsFormer successfully mitigates the failure modes for any combinations of $k\in [1e-2, 1e-3, 1e-4]$ and $\Delta t \in [5,7,10]$.

(2) The choice of $\Delta t$ should not be either too large (i.e., 1e-1) or too small (i.e., 1e-5). Intuitively, either a too-large or a too-small $\Delta t$ degrades the temporal dependencies between discrete time steps.

(3) Increasing the pseudo-sequence length can help mitigate PINNs failure modes (i.e., $k=3\rightarrow 5$ when $\Delta t=1e-4$). However, once PINNs successfully mitigate the failure mode, the benefit of further increasing $k$ is marginal.

---

### Meta-Review · Area_Chair_W9Qz · 2023-12-11

**Metareview:**

This paper presents to use transformers in PINNs to solve PDEs. The targeted approach is interesting and promising, as the original MLP-based PINNs may not have sufficient capacity in modeling complex spatial and temporal dependencies. While the basic idea of this work is straightforward, it seems to be the missing piece in the PINNs literature. So the paper is acceptable in this sense. However, reviewers held post-rebuttal concerns that the experimental evaluation is relatively weak, in that the considered PDEs were extensively studied with even better performance reported in recent works. Authors are urged to resolve these concerns by including more experimental results to solidify this work. Presentation quality can also be improved especially the figures shall be in high-resolution or vector format.

**Justification For Why Not Higher Score:**

A good idea that is missing in the PINNs literature but the empirical evaluation is decent rather than good. A poster is a proper decision.

**Justification For Why Not Lower Score:**

Reviewers are generally positive about this paper. Concerns from the negative reviewers were fairly addressed. The remaining concerns are relatively minor and can possibly be addressed in a minor revision.

---

### Decision · Program_Chairs · 2024-01-16

Accept (poster)